# Deep phenotyping of Alzheimer's disease leveraging electronic medical records identifies sex-specific clinical associations

Alice S. Tang [1,2,3✉], Tomiko Oskotsky [1,4], Shreyas Havaldar[5], William G. Mantyh[6], Mesude Bicak[5,7], Caroline Warly Solsberg [8,9,10], Sarah Woldemariam [1], Billy Zeng[3], Zicheng Hu [1], Boris Oskotsky [1], Dena Dubal [9], Isabel E. Allen[11], Benjamin S. Glicksberg [5,7] & Marina Sirota [1,4✉]

Alzheimer's Disease (AD) is a neurodegenerative disorder that is still not fully understood. Sex modifies AD vulnerability, but the reasons for this are largely unknown. We utilize two independent electronic medical record (EMR) systems across 44,288 patients to perform deep clinical phenotyping and network analysis to gain insight into clinical characteristics and sex-specific clinical associations in AD. Embeddings and network representation of patient diagnoses demonstrate greater comorbidity interactions in AD in comparison to matched controls. Enrichment analysis identifies multiple known and new diagnostic, medication, and lab result associations across the whole cohort and in a sex-stratified analysis. With this data-driven method of phenotyping, we can represent AD complexity and generate hypotheses of clinical factors that can be followed-up for further diagnostic and predictive analyses, mechanistic understanding, or drug repurposing and therapeutic approaches.

[1] Bakar Computational Health Sciences Institute, UCSF, San Francisco, CA, USA. [2] Graduate Program in Bioengineering, UCSF, San Francisco, CA, USA. [3] School of Medicine, UCSF, San Francisco, CA, USA. [4] Department of Pediatrics, UCSF, San Francisco, CA, USA. [5] Hasso Plattner Institute for Digital Health at Mount Sinai, Icahn School of Medicine at Mount Sinai, New York, NY, USA. [6] Department of Neurology, University of Minnesota School of Medicine, Minneapolis, MN, USA. [7] Department of Genetics and Genomic Sciences, Icahn School of Medicine at Mount Sinai, New York, NY, USA. [8] Pharmaceutical Sciences and Pharmacogenomics, UCSF, San Francisco, CA, USA. [9] Department of Neurology and Weill Institute for Neurosciences, University of California, San Francisco, San Francisco, CA 94158, USA. [10] Memory and Aging Center, UCSF, San Francisco, CA, USA. [11] Department of Epidemiology and Biostatistics, UCSF, San Francisco, CA, USA. ✉email: alice.tang@ucsf.edu; marina.sirota@ucsf.edu

Alzheimer's disease (AD) is the most common cause of dementia, making up 60–80% of cases, with a large and increasing burden on patients, caregivers, and society[1]. AD is characterized by brain atrophy and accumulation of beta-amyloid plaques and tau tangles seen on brain pathology after death. The disease erodes memory and cognitive functions, causing interference with daily activities and contributing to emotional, social, and economic burden on patients and their families. AD is incurable and challenging to understand and diagnose. One reason AD is difficult to study is because it is a complex, heterogeneous, and multifactorial disease that takes many years to manifest[2]. This complexity, along with the slow insidious progression of the disease, makes it difficult to fully characterize disease phenotypes and associations.

Sex is one factor that has been shown to be important in AD, with a higher prevalence in women afflicted by the disease at a 2:1 ratio compared to men[1]. While women have an increased estimated lifetime risk of AD, there is mixed evidence of risk between men and women of the same age[3,4]. Recent findings show that sex contributes to differing vulnerabilities or resilience to AD, as men with AD progress to death quicker[5,6] while women with this disease show higher cognitive resilience despite increased tau pathology[5,7,8]. How sex contributes to these differences in prevalence and vulnerability is a question of fervent interest among researchers in the AD field[9]. Recent studies in mice demonstrate that a second X chromosome may contribute to AD resilience[6]. Further sex-specific human studies in Alzheimer's disease also show sex modification of AD risk[10], progression[11], and molecular phenotype[11–15]. As such, sex is an important factor to consider in studying and phenotyping AD.

While many efforts have evaluated the association of individual risk factors with AD, unbiased approaches to these associations are limited. Prior work, largely hypothesis-driven, focused on select comorbidities associated with AD, such as hypertension[16], vascular disorders[17], diabetes[18], obesity[19], and others[20–22]. However, how sex modulates AD complexity and heterogeneity has still not been fully explored. Prior big data approaches to AD have examined genotype-phenotype associations[23,24] and molecular analyses[14,25–27] to characterize AD and sex differences[12,13]. Other work on phenotyping patients with AD using clinical data has examined neuroimaging[28], neuropsychiatric phenotype[29], chart reviews[30], and billing records independently. Thus, an unbiased comprehensive approach to phenotype AD and identify sex associations using full clinical records is needed.

With the rise in electronic medical record (EMR) use over the past decade[31], there is abundant underutilized clinical data on patients covering comorbidities, medications, and lab values. This type of data set provides a great opportunity to deeply investigate diseases and identify associations to facilitate understanding disease prevention and progression. Recently, EMR has been utilized for other diseases for creating comorbidity networks[32], identifying disease subtypes[33] and predicting disease outcomes[34,35] highlighting the potential of utilizing EMR data to extract insight and utility for complex and heterogeneous diseases[36], but a big data integrative analysis with EMR data has not yet been applied to characterize AD.

Deep phenotyping is a data-driven approach that has been used to provide more detailed stratification and representation of a disease in the era of precision medicine[37,38]. Here, we take an integrative approach through deep clinical phenotyping and network analysis to provide insight into AD clinical characteristics with a focus on sex differences. For the first time to our awareness, integrative phenotyping and association analysis is used to identify, in an unbiased manner, unique clinical features associated with AD itself—and reveals potential previously unknown sex-specific associations in the context of diagnoses, medications, and lab test results.

## Results

From the UCSF EMR database (~5 million patients), we identified 8804 patients with AD (5558 female, 86.5 mean age (6.4 standard deviation)) and 17,608 propensity score (PS)-matched control patients (11,117 females, 86.5 mean age (6.4 standard deviation)). From the Mount Sinai EMR (~4 million patients), 5958 patients with AD (4138 females, 88.3 mean age (8.7 standard deviation)) and 11,916 PS-matched controls (8446 females, 88.7 mean age (11.4 standard deviation)) were identified (Fig. 1). Male and female groups were identified by the most recent sex assignment in the EMR, and race/ethnicity information was extracted from the EMR as reported by the patient. Post-matching analysis demonstrated the adequate balance in covariates with standardized mean differences in age and categorical distributions below 0.1 (or below 0.2 between matched sex groups). Demographic characteristics of patients with AD and matched control patients are shown in Table 1 and Supplementary Table 1.

**Embedding with diagnosis shows separation between AD and controls.** Due to the size of our cohort, we first performed low-dimensional visualizations using diagnoses as features to visualize patient separation. Low-dimensional UMAP visualizations of non-AD diagnoses (47,439 features, ICD-10-CM codes) show that distributions for patients with AD and controls are significantly different among the first two UMAP components (two-sided Mann–Whitney $U$-test, $p$-value < 1e−5, Fig. 2a, b) at both UCSF and Mount Sinai, with a progressive separation between groups. For the UCSF data, sex, and death status show significant correlations with the first component, while age is significantly correlated with both components (two-sided Mann–Whitney $U$-test $p$-value < 0.01, Fig. 2a, Supplementary Fig. 1). Sex, death status, and age are significantly correlated with both components at Mount Sinai (two-sided Mann–Whitney $U$-test $p$-value < 0.01, Fig. 2b, Supplementary Fig. 1).

**Association analysis identifies associated comorbidities in AD.** Among each diagnostic hierarchical level (Level 2 categories, Level 3 categories, and full diagnosis names), the majority of AD disease networks contain more nodes and edges compared with control networks (Supplementary Table 3). In UCSF Level 3 diagnosis networks, more nodes and edges occur in AD vs control networks. As shown in Fig. 3a, when thresholding Level 3 diagnosis categories by >10% of patients, there are 144 diagnosis pairs among patients with AD compared to one pair in controls. When comparing node-level network metrics between groups, thresholded by >1% of patients within a group, AD and control networks are significantly different when compared on closeness centrality, degree, neighborhood connectivity, and stress centrality indicating a higher degree of connectivity among AD networks across all levels (two-sided Mann–Whitney $U$-test, $p$-value < 0.01, Fig. 3c). In Mount Sinai Level 3 diagnostic networks, more edges occur in AD networks compared to control networks, with significantly different distributions across AD and control networks on degree, neighborhood connectivity, and stress centrality (two-sided Mann–Whitney $U$-test, $p$-value < 0.01, Supplementary Table 3). Across the board, network metrics normalized by the metric are significantly correlated between UCSF and Mount Sinai (Spearman's $\rho = 0.44$, $p$-value < 1e−4, Fig. 3e).

Within Level 2 diagnosis categories, there are 166 significant diagnosis categories (two-sided Fisher's exact or Chi-squared test, Bonferroni-corrected $p$-value < 0.05), with 120 diagnosis categories significantly enriched (odds ratio (OR) > 2) uniquely in the AD group and no significantly enriched diagnosis categories uniquely in the control group (Fig. 4a, top). Within Level 3 diagnosis categories,

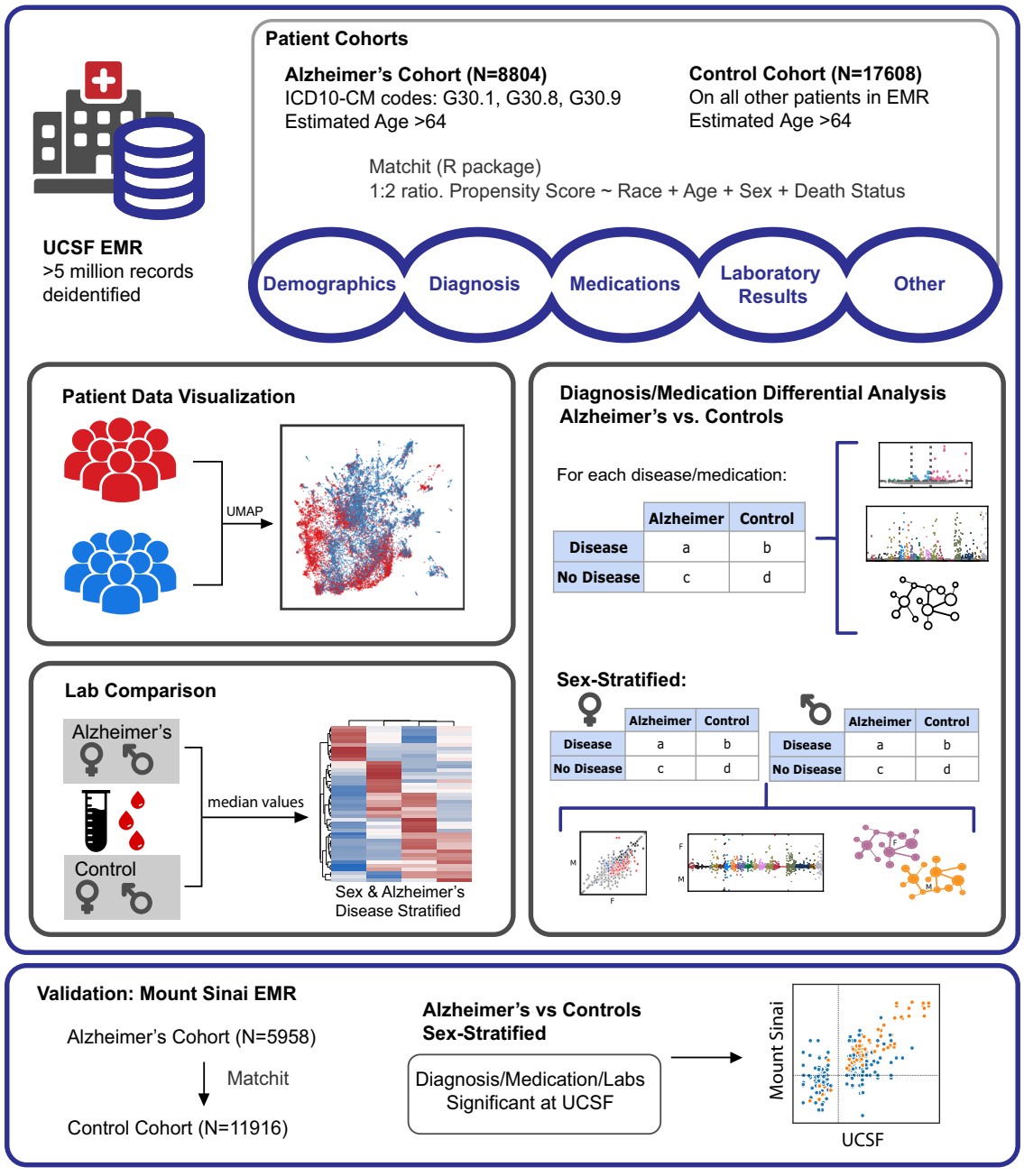

**Fig. 1 Workflow visualization.** Visualization of patient cohort identification from the UCSF EMR and methods for deep phenotyping and enrichment analysis. Validation analysis is done with Mount Sinai EMR to assess correlations.

there are 501 significant categories, with 391 and 4 categories significantly enriched in AD and control groups, respectively (two-sided Fisher exact or Chi-squared test, Bonferroni-corrected $p$-value < 0.05, Supplementary Table 2). Within full diagnosis names, there are 1627 significant diagnoses, with 1491 and 7 diagnoses enriched uniquely in AD and control groups, respectively. Top significant diagnoses in AD include vascular dementia, hypertension, hyperlipidemia, urinary tract infection, syncope, hypothyroidism, and osteoporosis, while top significant diagnoses in controls include neoplasms of liver and brain (two-sided Fisher exact or Chi-squared test, Bonferroni-corrected $p$-value < 0.05, Fig. 4a, bottom,Supplementary Data 1). Top ICD diagnostic blocks in AD include mental health and behavioral diseases, genitourinary diseases, endocrine and metabolic diseases, and circulatory system diseases (Fig. 4b). In the validation cohort, 1495 of 1627 significant UCSF diagnoses mapped to Mount Sinai EMR codes, of which 889

(60.13%) are significant (two-sided Fisher's exact or Chi-squared test, Bonferroni $p$-value < 0.05). Overall comorbidity odds ratios at UCSF are significantly correlated with those of the validation cohort at Mount Sinai (Spearman ρ = 0.65, $p$-value < 1e−5, Fig. 4c).

**Sex-stratified AD vs. control association analysis identifies vascular and musculoskeletal disorders in females with AD and behavioral/neurological disorders in male AD.** When stratifying diagnoses by sex (see "Methods" section), AD disease networks are significantly different on metrics of degree and neighborhood connectivity in both males and females compared to their respective controls among all diagnostic hierarchical levels ($p$-value < 0.001). Comparison of sex-specific AD network for diagnosis name shows significantly greater neighborhood connectivity, and lower eccentricity in female networks (two-sided

**Table 1 Patient demographics.**

| | UCSF | | | | Mount Sinai | | | |
|---|---|---|---|---|---|---|---|---|
| | Overall | AD | Control | SMD | Overall | AD | Control | SMD |
| *n* | 26,412 | 8804 | 17,608 | | 17,874 | 5958 | 11,916 | |
| Sex, *n* (%) | | | | | | | | |
| Female | 16,675 (63.1) | 5558 (63.1) | 11,117 (63.1) | <0.001 | 12,584 (70.4) | 4138 (69.5) | 8446 (70.9) | 0.031 |
| Male | 9659 (36.6) | 3220 (36.6) | 6439 (36.6) | | 5290 (29.6) | 1820 (30.5) | 3470 (29.1) | |
| Unknown | 78 (0.3) | 26 (0.3) | 52 (0.3) | | | | | |
| Estimated age, mean (SD) | 86.5 (6.4) | 86.5 (6.4) | 86.5 (6.4) | <0.001 | 88.6 (10.6) | 88.3 (8.7) | 88.7 (11.4) | −0.039 |
| Race, *n* (%) | | | | | | | | |
| American Indian/ Alaska Native | 27 (0.1) | 9 (0.1) | 18 (0.1) | <0.001 | 20 (0.1) | 8 (0.1) | 12 (0.1) | 0.129 |
| Asian | 2638 (10.3) | 879 (10.3) | 1759 (10.3) | | 177 (1.0) | 78 (1.3) | 99 (0.8) | |
| Black/African American | 1758 (6.9) | 586 (6.9) | 1172 (6.9) | | 3732 (20.9) | 1214 (20.4) | 2518 (21.1) | |
| Native Hawaiian/ Pacific Islander | 1356 (5.3) | 452 (5.3) | 904 (5.3) | | 9 (0.1) | 5 (0.1) | 4 (0.0) | |
| Other | 2230 (8.7) | 743 (8.7) | 1487 (8.7) | | 3922 (21.9) | 1496 (25.1) | 2426 (20.4) | |
| Unknown | 2017 (7.6) | 673 (7.6) | 1344 (7.6) | | 786 (4.4) | 253 (4.2) | 533 (4.5) | |
| White/Caucasian | 16,386 (64.0) | 5462 (64.0) | 10,924 (64.0) | | 9228 (51.6) | 2904 (48.7) | 6324 (53.1) | |
| Death status, *n* (%) | | | | | | | | |
| Alive | 20,146 (76.3) | 6714 (76.3) | 13,432 (76.3) | 0.001 | 9371 (52.4) | 3264 (54.8) | 6107 (51.3) | 0.078 |
| Deceased | 6266 (23.7) | 2090 (23.7) | 4176 (23.7) | | 882 (4.9) | 306 (5.1) | 576 (4.8) | |
| Unknown | | | | | 7621 (42.6) | 2388 (40.1) | 5233 (43.9) | |

Summary table of sex, estimated age, death status, and first race among Alzheimer's and control cohorts at UCSF and Mount Sinai. Patients are propensity score-matched at a 1:2 Alzheimer to control ratio with the demographics shown in the table.
*SD* standard deviation, *SMD* standardized mean difference.

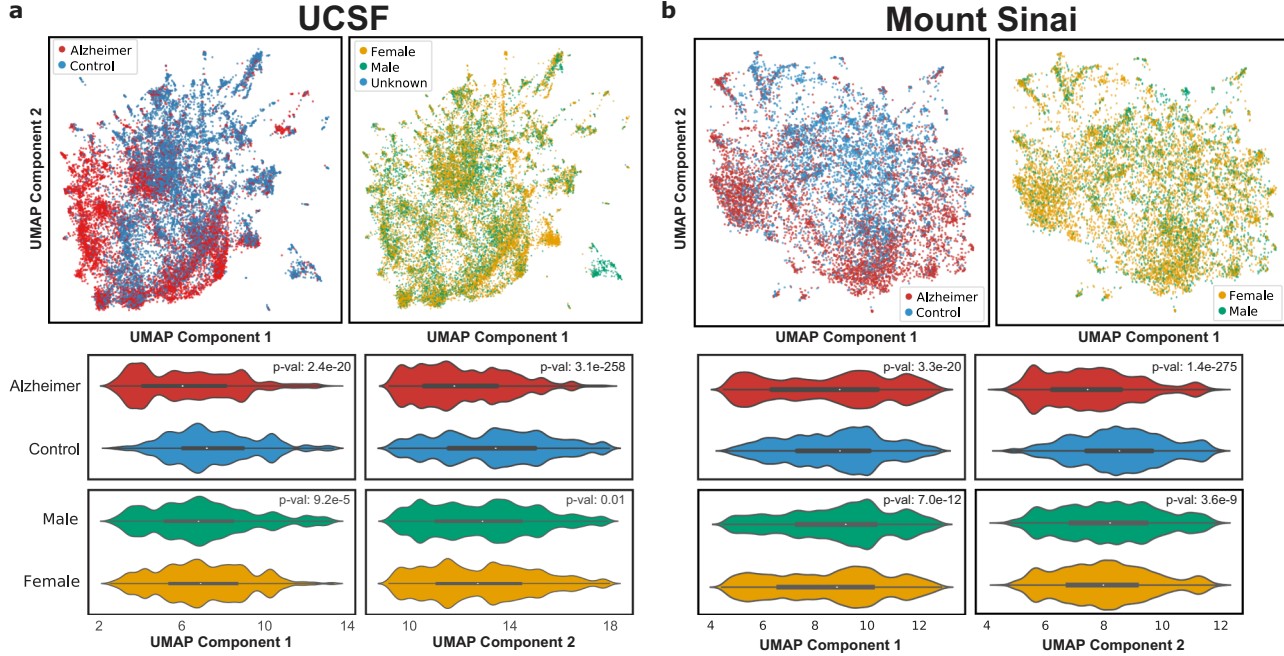

**Fig. 2 UMAPs using comorbidities as features provide a topographical view of the distribution of patients.** Top row: UMAP of all patients (AD and controls), with each dot representing a patient, colored by AD status (**a** top left, **b** top left) or by sex (**a** top right, **b** top right). Middle and bottom rows: violin plots show the distribution of patients with AD and controls along the UMAP principal components for UCSF (**a**) and Mount Sinai (**b**), and *p*-values determined from comparing distributions with a two-sided Mann–Whitney *U*-test. Alzheimer vs Control: UCSF *p*-value 2.4e−20 (component 1) and 3.1e−258 (component 2). Mount Sinai *p*-value 3.3e−20 and 1.4e−275. Male vs female: UCSF *p*-value 9.2e−5 and 0.01. Mount Sinai *p*-value 7.0e−12 and 3.6e−9.

Mann–Whitney *U*-test, *p*-value < 0.01 both metrics, Fig. 3d, Supplementary Table 3). Within the validation cohort, similarly, female AD networks show significantly greater neighborhood connectivity compared to male AD networks (two-sided Mann–Whitney *U*-test, *p*-value < 0.01, Supplementary Table 3).

When thresholding full diagnosis names by >10% of patients within a sex group, female patients with AD have 58 shared co-diagnosis pairs compared to 38 in male patients with AD (Fig. 3b and Supplementary Table 3), and 3 shared co-diagnosis pairs were identified for both control sex groups.

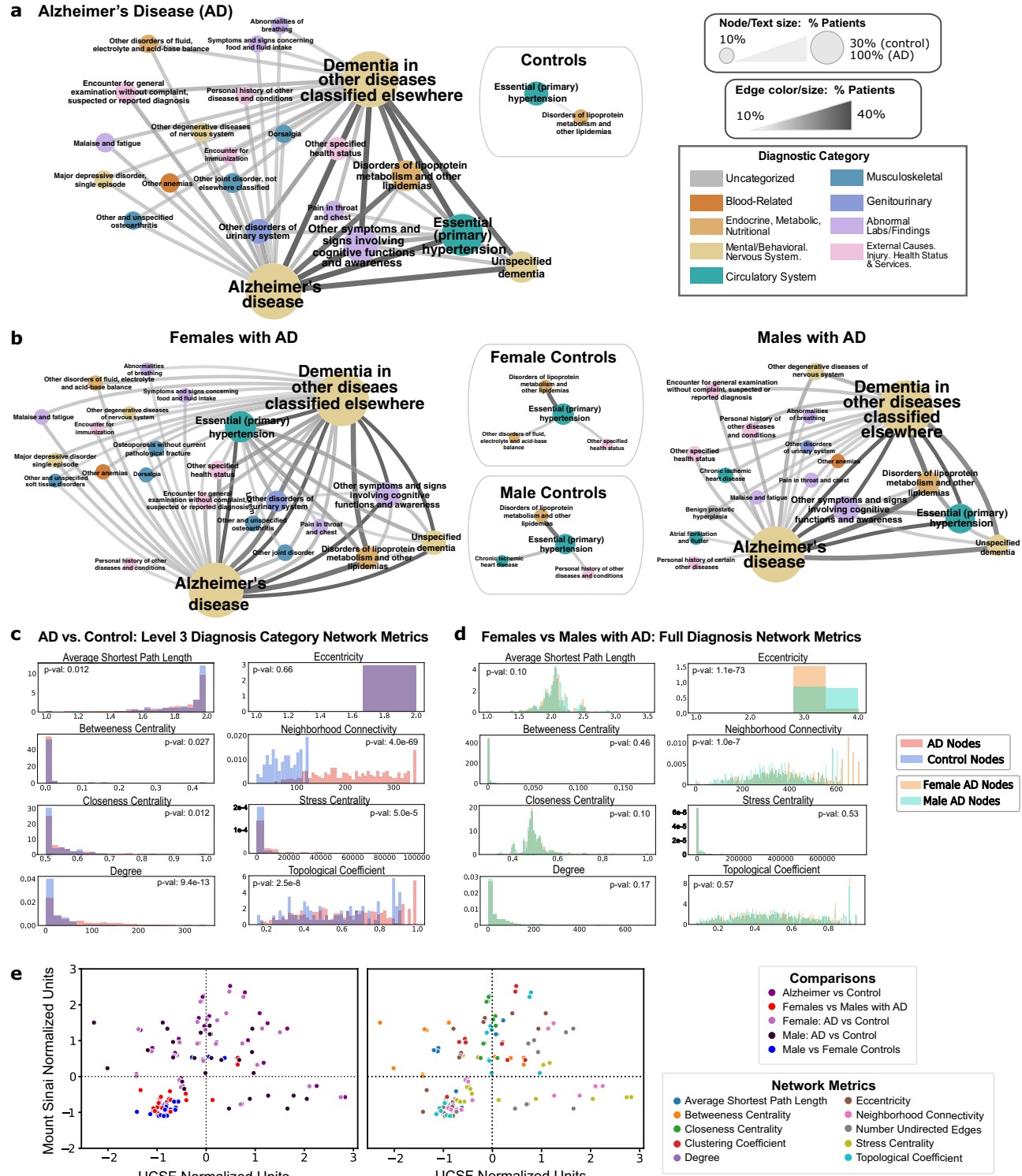

**Fig. 3 Comorbidity networks show greater co-diagnosis in patients with AD vs. controls, and in females with AD vs males with AD. a, b** Network diagrams: For each network, the node size, text size, edge size, and edge color represent the number of patients sharing a diagnosis or diagnosis pair. Node colors are based on ICD-10-CM category. A threshold of 10% sharing was applied. **a** Network for Level 3 diagnosis categories in patients with AD vs. controls. Nodes and edges represent >10% of diagnosis or diagnosis pairs shared in each cohort, respectively. **b** Female and male network of Level 3 diagnosis categories for patients with AD and controls. Each node and edge represent a diagnosis or diagnosis pairs shared by >10% of males or females in the AD or control group. **c** Comparison of Level 3 diagnosis category network metrics between patients with AD and controls. Statistical tests are performed with a two-sided Mann–Whitney U-test. Significant metrics with p-value < 0.01: degree (9.4e−13), neighborhood connectivity (4.0e−69), stress centrality (5.0e−5), and topological coefficient (2.5e−8). **d** Comparison of network metrics between male and female Alzheimer's disease full diagnostic name networks. Statistical tests are performed with a two-sided Mann–Whitney U-test. Significant metrics with p-value < 0.01: eccentricity (1.1e−73) and neighborhood connectivity (1.0e−7). **e** Correlation of network metrics compared with validation EMR network metrics, normalized by the metric. Colors represent comparison type (left) or the specific network metric (right), Spearman's ρ = 0.55, p-value < 1e−4.

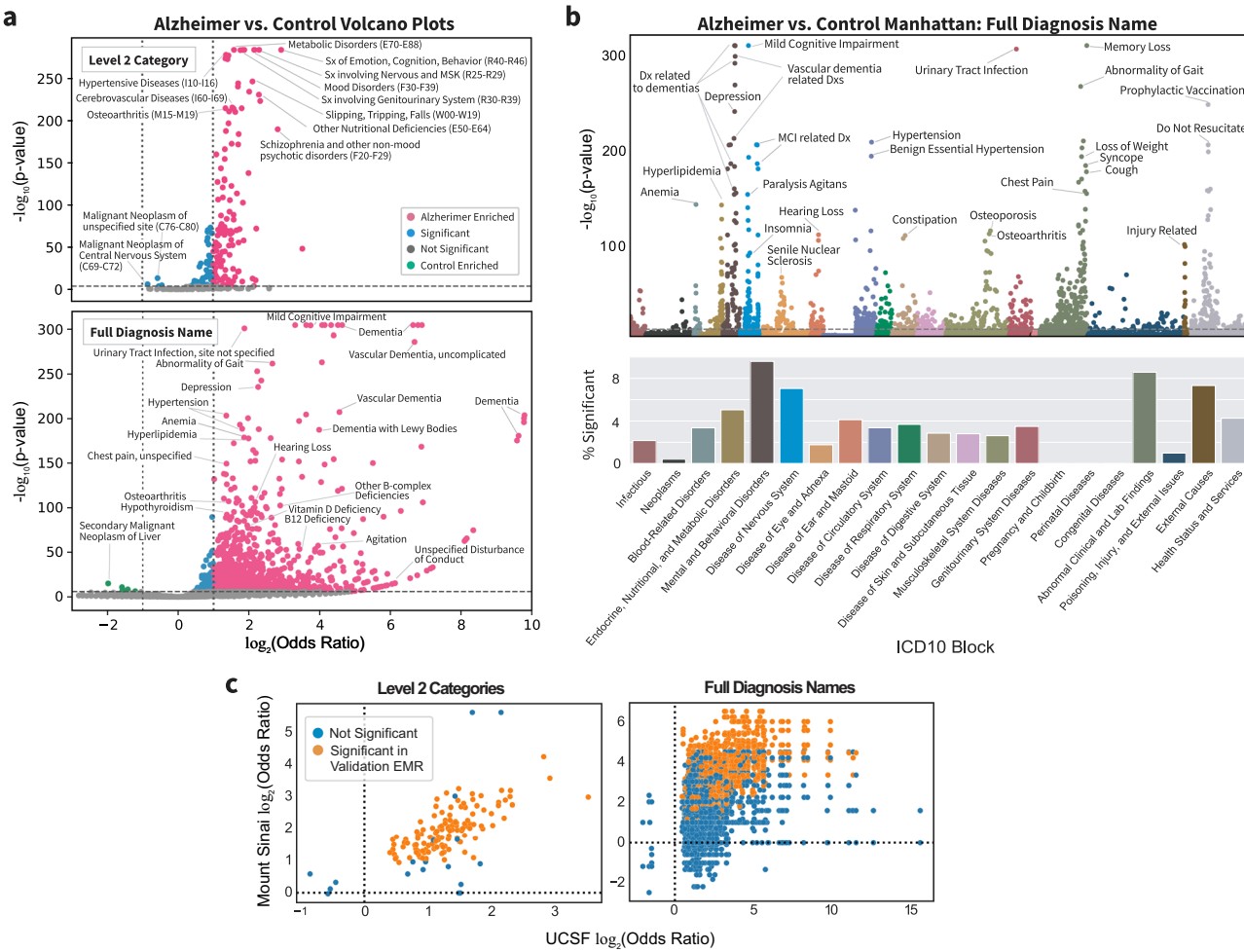

**Fig. 4 Comorbidity enrichment analysis identifies enriched diagnosis in AD vs. control cohorts. a** Volcano plot for Level 2 categories (top) and full diagnosis names (bottom) compared between AD and control cohorts using two-sided Fisher's exact or Chi-squared test. *p*-value cutoff is Bonferroni-corrected (*p*-value < 2e−8 and 1e−6) with log2 odds ratio cutoff of 1 for AD-enriched (pink) or log2 odds ratio cutoff of −1 for control-enriched (green) and remaining significant diagnoses in blue. Some of the top significant diagnoses are labeled. **b** Top, a Manhattan plot with full diagnosis names colored by ICD-10-CM categories with significance determined by two-sided Fisher's exact or Chi-squared test with Bonferroni-corrected *p*-value threshold of 0.05. Some of the top diagnoses in each category are labeled. Bottom, the percentage of diagnosis in each ICD-10-CM category is significant. **c** Diagnosis AD vs. control odds ratio correlation plots between UCSF and Mount Sinai for Level 2 diagnosis categories and full diagnosis names that are significant at UCSF (two-sided Fisher's exact or Chi-squared test, Bonferroni-corrected *p*-value threshold of 0.05). Each dot represents a category or diagnosis, and dots in orange are significant at Mount Sinai with (two-sided Fisher's exact or Chi-squared test with Bonferroni-corrected *p*-value threshold of 0.05 based on the number of significant UCSF diagnoses).

For both males and females, there are 136, 338, and 714 shared significant diagnostic categories or diagnoses for Level 2, Level 3, and full diagnosis names, respectively. In a sex-stratified analysis, there were 29, 164, and 699 female-only significant hits and 5, 18, and 91 male-only significant hits for Level 2, Level 3, and full diagnosis names (two-sided Fisher's exact or Chi-squared test, Bonferroni-corrected *p*-value < 0.05, Fig. 5a, Supplementary Data 1). Compared to males among Level 2 diagnostic categories, females have a greater percent of significant diagnoses in blood-related disorders (e.g., nutritional anemia, coagulation defects) and congenital disorders and also have greater enrichment of pervasive and specific developmental disorders, musculoskeletal disorders (e.g., chondropathies, other osteopathies), injuries (e.g., injuries to the hip and thigh, injuries to the ankle and foot), infections with a predominantly sexual mode of transmission, and metabolic disorders (Supplementary Data 1). When comparing Level 2 categories in the validation cohort, among females, 153 out of 165 mapped with 60 (30.22%) significant, and among

males, 133 out of 141 mapped with 64 (48.12%) significant (two-sided Fisher's exact or Chi-squared test, Bonferroni-corrected *p*-value < 0.05 based on the number of significant UCSF diagnoses). In general, Level 2 category sex-specific odds ratios are correlated between institutions (Females: Spearman's ρ = 0.77, *p*-value < 1e−5; males: Spearman's ρ = 0.83, *p*-value < 1e−5). In the validation cohort, females have similar enrichment of blood-related disorders (e.g., nutritional anemia) and injuries (e.g., injuries to the hip and thigh), while males have enrichment of behavioral/emotional disorders.

Within full diagnosis names, unique significant diagnoses of female patients with AD include asthma, atrial fibrillation, arthritis, fractures, and accidents while unique significant diagnoses of male patients with AD include parkinsonism, sleep apnea, hypersomnia, neuropathy, irritability, and imbalance (two-sided Fisher's Exact or Chi-squared test, Bonferroni-corrected *p*-value < 0.05, Fig. 5a, b, Supplementary Data 1). Among full diagnosis names significant in both males and

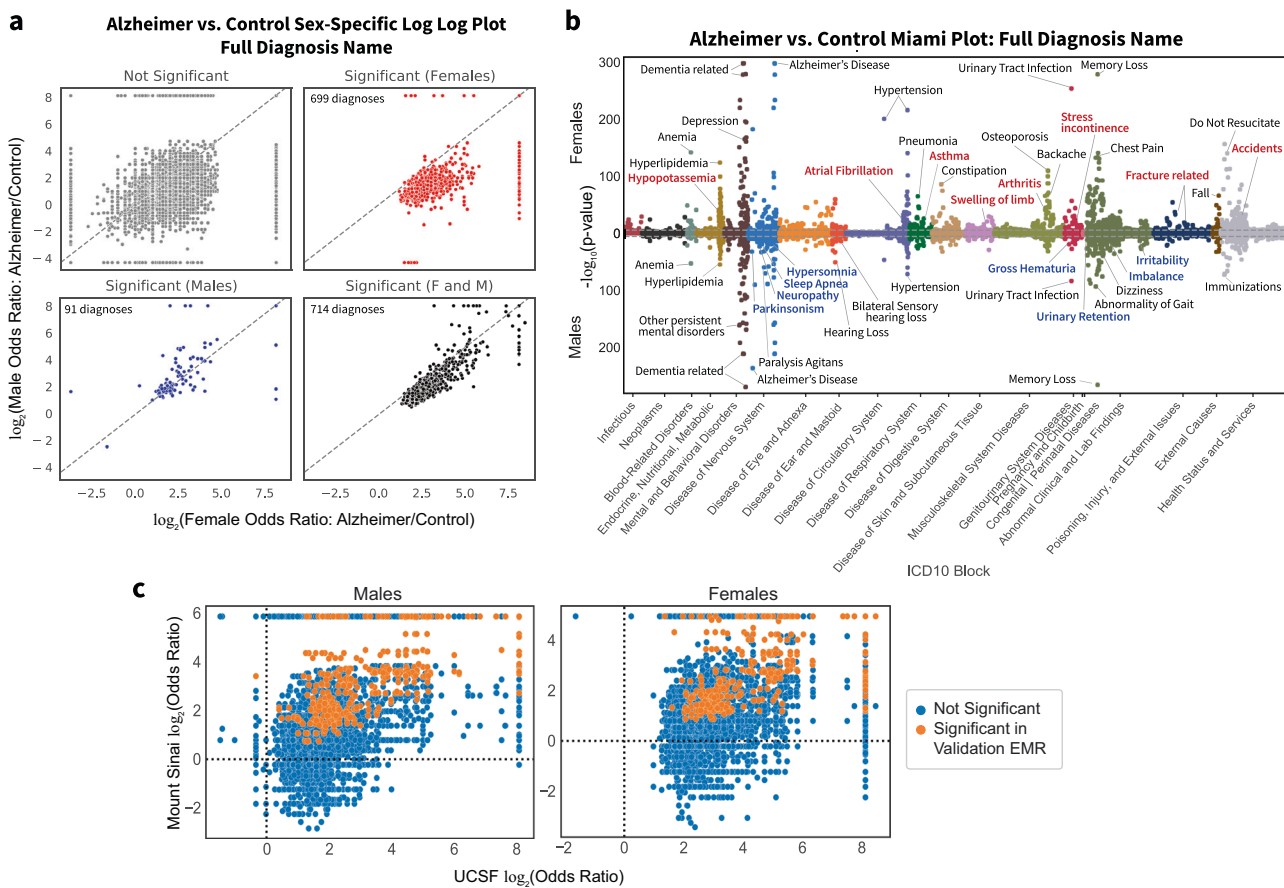

**Fig. 5 Comorbidity enrichment analysis identifies sex-specific enriched diagnoses in AD vs. control cohorts. a** Full diagnosis names compared between patients with AD and controls within each sex. The log2 of the odds ratio for each sex is plotted on the axis, and points are colored by significance (Bonferroni-corrected, *p*-value cutoff > 3e−6). **b** Miami plot of the diagnosis names grouped by sex and ICD-10-CM categories. Select top diagnoses are labeled, with diagnosis names colored by significance as female-only (red), male-only (blue), or significant in both sexes (black). **c** Correlation plots of AD vs. control odds ratios between UCSF and Mount Sinai for diagnoses that are significant at UCSF for each sex group (two-sided Fisher's exact or Chi-squared test, Bonferroni-corrected *p*-value threshold of 0.05). Each dot represents a diagnosis, and dots in orange are significant at Mount Sinai (two-sided Fisher's exact or Chi-squared test with Bonferroni-corrected *p*-value threshold of 0.05 based on the number of significant UCSF diagnoses for each sex group).

females, female patients with AD have a greater association in depression, hypertension, hyperlipidemia, urinary tract infections, upper respiratory infections, anemia, osteoporosis, and pneumonia, while male patients with AD have greater effect size with behavioral phenotypes, hearing loss, and agitation (Supplementary Data 1). Among the full diagnosis names in the validation cohort, for females, 1149 out of 1383 significant diagnoses mapped, of which 240 (20.89%) were significant, and for males, 702 out of 805 significant diagnoses mapped, of which 216 (30.77%) were significant. In general, sex-specific diagnosis odds ratios were correlated for both females (Spearman's ρ = 0.77, *p*-value < 1e−4) and males (Spearman's ρ = 0.83, *p*-value < 1e−4, Fig. 5c). In the validation cohort, similarly, female patients with AD have a greater association in depression, hypertension, and osteoporosis while male patients with AD have a greater association in hearing loss and agitation (Supplementary Data 1).

**Few comorbidities change with sensitivity analysis on encounters**. For our sensitivity analysis that included only patients with ≥10 encounters (i.e., recorded outpatient, inpatient, or emergency room visits to UCSF Health) in EMR and with visits spanning >1 year, there were 6612 patients with AD (2382 males, 4223 females) and 13,224 control patients (4674 males,

8539 females) identified by PS-matching on the number and timespan of encounters in addition to demographic characteristics and death status. A summary of the demographic characteristics of these cohorts is shown in Supplementary Table 1. We identified 100, 222, and 561 significant Level 2, Level 3, and full diagnosis names respectively (two-sided Fisher's exact or Chi-squared test, Bonferroni-corrected *p*-value threshold of 0.05), and an increase in the odds ratio for chromosomal abnormalities and cerebrovascular disorders in patients with AD (Supplementary Data 2). With sex-stratified enrichment analysis, encounter controlling increased enrichment of cerebrovascular disease in females, and increased significant enrichment of behavioral disorders, vision problems, and vascular dementia in males (Supplementary Data 2). An interactive visualization of Figs. 3 and 4 is made available in an Rshiny app vizad.org.

**Medication association analysis identifies dexamethasone as enriched in controls**. In addition to comorbidities, we performed medication enrichment analysis in order to phenotype patients and investigate medication prescriptions enriched in patients with AD and controls. Medications found enriched (two-sided Fisher's exact or Chi-squared test, Bonferroni-corrected *p*-value < 0.05, OR > 2 or < 0.5) in patients with AD include current treatments

like donepezil and memantine, but also vitamin B12, anti-depressants (escitalopram, citalopram, sertraline, mirtazapine, trazodone), antipsychotics (quetiapine, risperidone, olanzapine), carbidopa/levodopa, vitamin D3, and melatonin. Medications found enriched in control patients include dexamethasone, ondansetron, and alteplase. Significant medications in controls with lesser effect size (two-sided Fisher's exact or Chi-squared test, Bonferroni-corrected $p$-value < 0.05, 0.5 < OR < 1) include midazolam, propofol, opioids (oxycodone, fentanyl citrate), and furosemide (Fig. 6a). From the validation cohort, 116 out of 121 medications mapped, of which 66 (56.90%) were significant (two-sided Fisher's Exact or Chi-Squared test, Bonferroni-corrected $p$-value < 0.05 based upon significant medications at UCSF). In general, odds ratios of medications are significantly correlated (Spearman's $\rho = 0.85$, $p$-value < 1e−4, Fig. 6c). Dexamethasone is significant among controls in both institutions, and multiple medications including vitamin B12, antidepressants, and antipsychotics are significant in patients with AD among both institutions.

In a sex-stratified analysis, medications enriched in males with AD include Tdap vaccine, melatonin, and carbidopa/levodopa while methylprednisolone and phenylephrine are enriched in control males. Female patients with AD have enrichments in diazepam, antipsychotics (risperidone, aripiprazole), buspirone, antidepressants (sertraline, mirtazapine, trazodone, bupropion), vitamin D2, and levothyroxine while control females are enriched in norepinephrine bitartrate and fentanyl citrate (Fig. 6b). In the validation EMR, 18 of 23 (78.25%) significant medications found at UCSF are significant in females at Mount Sinai, and 13 of 16 (81.25%) in males (two-sided Fisher's exact or Chi-squared test, Bonferroni-corrected $p$-value < 0.05 based upon significant medications at UCSF within a group). Overall, there is significant correlation of sex-specific medication odds ratios in females (Spearman's $\rho = 0.7$, $p$-value = 0.001) and males (Spearman's $\rho = 0.62$, $p$-value = 0.001, Fig. 6c). Among both institutions, carbidopa/levodopa is significant in males with AD only.

**Comparing labs between sex-specific AD and control groups identifies clusters of lab value differences.** We also performed an unbiased analysis of laboratory test result differences between patients with AD and controls to phenotype patient groups. Among significantly different median lab values in both UCSF and Mount Sinai, patients with AD have higher levels of hematocrit, serum calcium, RBC count, serum albumin, and cholesterol and lower levels of glucose, activated partial thromboplastin time (aPTT), alanine transaminase (ALT), and aspartate transaminase (AST) compared to controls (two-sided Mann–Whitney $U$-test, Bonferroni-corrected $p$-value threshold of 0.05, Fig. 6d, Supplementary Fig. 4A).

Average significant median lab values across sex-stratified groups (females with AD, males with AD, control females, control males) and across institutions were clustered into 7 significant clusters (Family-wise Error Rate (FWER) corrected $p$-value 0.05 cutoff, Fig. 6d). Clusters 1, 4, and 7 show discordant results between UCSF and Mount Sinai. Cluster 2 represents groups of significant median lab values lowest in control males, and highest either in all patients with AD (e.g., albumin, sodium, and carbon dioxide) or highest in females with AD (e.g., HDL cholesterol, lymphocytes, calcium). Cluster 3 represents significant labs with greater median values in females and in controls (e.g., Free T4, sedimentation rate). Cluster 5 represents labs with lower significant median values in patients with AD than controls for either the whole group (e.g., B-Type Natriuretic Peptide, AST) or in a sex-specific way where significant median lab values for males are greater than for females (e.g., aPTT, ALT, ferritin). Cluster 6 shows labs greater in AD compared to controls in a sex-specific way where overall males have greater significant median lab values than females (e.g., hemoglobin, RBC count). Across the board, the normalized lab values are correlated between the institutions (Female control: Spearman's $\rho = 0.45$, $p$-value < 0.001; male control: 0.46, $p$-value < 0.001; female AD: 0.59, $p$-value < 1e−5; Male AD: 0.64, $p$-value < 1e−5; Supplementary Fig. 4B).

**Discussion**

In this work, we demonstrate the capability of utilizing data from EMRs in order to perform deep phenotyping of a complex and heterogeneous disease, Alzheimer's Disease (AD), and derive insights into associations with AD in a combined and sex-stratified analysis.

First, we performed low-dimensional topographical embedding of patients using diagnoses as features in order to visualize patients spatially. We see that AD status is significantly correlated with the first two UMAP components at both institutions, suggesting that phenotypic representation of patients using diagnosis data can demonstrate separation of patients with AD and control patients. The UMAP representation demonstrates a progressive spectrum between control patients and patients with AD, as well as representing variance and heterogeneity at individual patient resolution. Furthermore, with the UMAP representation, we can visualize topographically the distribution of age, sex, and other variables among patients.

We then generated comorbidity networks between patients with AD and control patients which provide a phenotypic representation of disease interactions among patient groups and a difference in connectivity between diseases in patients with AD and control patients. AD networks contain a greater number of edges and network metrics that point to higher rates of comorbid conditions among patients with AD at both institutions, particularly with stronger links of hypertension (HTN)—lipidemias and HTN—urinary disorders. Indeed, other studies have found multimorbidities (such as neuropsychiatric and cardiovascular patterns) to increase the risk for dementia[39], and to contribute to AD pathological heterogeneity[40,41] displaying the larger complexity and heterogeneous nature of AD.

With enrichment analysis, we applied an integrative, unbiased, big data approach to EMR and identified previously known associations and possible novel connections with AD. Some diagnoses found enriched in patients with AD compared to control patients from our analysis at both institutions that have been previously identified as linked with AD include midlife hypertension[16,42], diabetes mellitus[18,43], anemia[44,45], vascular pathology[17,46], osteoporosis[47,48], and urinary tract infections (UTI)[49]. Enrichment of hypertension and vascular risk factors supports many current hypotheses of potential vascular pathologies and inflammatory factors that may lead to AD[17,50–52] or "unmask" the symptoms of AD by decreasing cognitive reserve by causing vascular brain disease. Enrichment of diabetes and dyslipidemia supports existing literature that found links with diabetes mellitus and dyslipidemia[53], with proposed hypotheses involving energy metabolism[54–56], inflammation[57–59], or the integrity of the blood–brain barrier[60–62]. Enrichment of degenerative diseases of age, such as osteoporosis, osteoarthritis, urinary issues, and sensory issues may align with theories of AD as being a disease linked with frailty[63–65]. This analysis, therefore, provides an unbiased integrative way to identify multifactorial associations with AD. Our enrichment analysis also identified neoplasms as enriched in controls at UCSF, especially cancer of the brain and liver. While this is an associative finding, this supports ideas that cancer and AD co-occur less

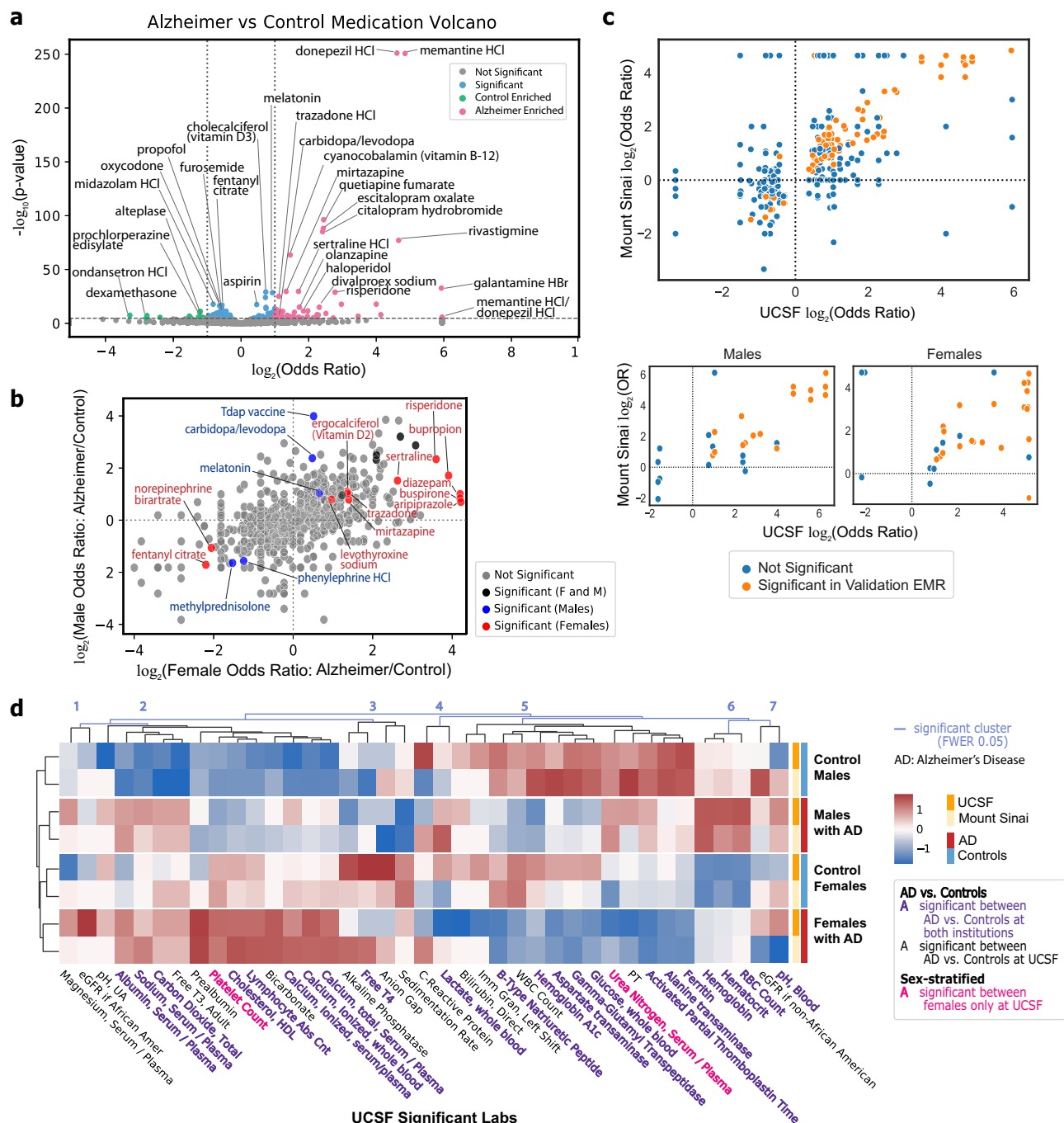

**Fig. 6 Medication and lab analysis shows medication enrichments and median lab value differences between AD and control cohorts. a** Volcano plot for generic medication names compared between AD and control cohorts using two-sided Fisher's exact or Chi-squared test. *p*-value cutoff is Bonferroni-corrected (*p*-value < 2e−5) with odds ratio cutoff at 2 for AD-enriched (pink) or 1/2 for control-enriched (green). Remaining significant diagnoses are in blue. **b** Log–log plot of generic medication names compared between AD and control cohorts within each sex. The log of the odds ratio for each sex is plotted on the axis, with points colored by significance based upon two-sided Fisher's exact or Chi-squared test with Bonferroni-corrected threshold of 0.05 if female-only (red), male-only (blue), or both (black). **c** AD vs control (top) and sex-specific (bottom) odds ratio correlation plots between UCSF and Mount Sinai for medications significant at UCSF (two-sided Fisher's exact or Chi-squared test with Bonferroni-corrected *p*-value threshold of 0.05). Each dot represents a medication, and the dots in orange are significant at Mount Sinai (two-sided Fisher's exact or Chi-squared test with Bonferroni-corrected *p*-value threshold of 0.05 based on the number of significant UCSF diagnoses in each group). **d** Heatmap of lab values filtered on significance at UCSF in AD vs control comparison across sex-specific groups at UCSF and Mount Sinai. Labs are clustered with light blue lines representing significant cluster breaks (family-wise error rate (FWER)-corrected *p*-value 0.05). Text color represents significant labs at both institutions (purple), significant among females only at UCSF (red), or significant between AD vs controls at UCSF only (black). Heatmap colors represent *z*-score of the average median value across the 4 groups at each institution.

frequently than the general population[66,67]. Some theories propose that AD and cancer have similar mechanisms and molecular pathways, but are dysregulated in different directions[68,69].

Next, we generated sex-specific comorbidity networks to provide insight into sex differences in the complexity of the disease. In both EMRs, female AD networks contain more nodes with network metrics suggesting greater connectivity than female controls or male AD networks. This may support the association with greater combined diagnoses and multimorbidity in female patients with AD compared to males[70]. These associations would be consistent with theories of greater risk of dementia in females as a result of multiple diseases or the theory of greater cognitive and pathological resilience to AD in females due to the burden of more comorbidities. Furthermore, sex-stratified networks show secondary interactions between comorbidities and AD, such as links of HTN-UTI and HTN-chest pain among female AD populations, but not in male patients with AD. These findings give higher-order comorbidity interactions associated with AD that have not been examined previously.

When performing enrichment analysis, we identify sex-specific enrichments that may be linked to AD that have not been previously explored in depth. Male patients with AD show enrichment of neurological and sensory disorders (sleep disorder, parkinsonism, and irritability), and among diagnoses significant in both sexes, males with AD have a stronger effect size with behavioral diagnoses, agitation, and hearing loss. These disorders are also mostly shown to be significant and associated with greater effect size compared to females in our validation cohort. Prior studies have found hearing loss to increase risk of dementia diagnosis[71,72] or cognitive decline[73,74] in men. The enrichment of behavioral and neurological disorders found in male patients with AD may indicate lessened resilience or higher occurrence of co-pathology. Furthermore, this analysis found the psychiatric phenotype associated with AD to be related to behavioral phenotypes in males compared to females, which is consistent with prior studies[75,76].

Female patients with AD have enrichment of unique significant diagnoses in musculoskeletal categories (arthritis, fractures), atrial fibrillation, and accidents, and among diagnoses significant in both sexes, females with AD show stronger effect size with depression, hypertension, urinary tract infections, and osteoporosis. Some of these disorders are similarly significant and associated with greater effect sizes compared to males in our validation cohort. The diagnoses of hypertension and atrial fibrillation would be in line with the hypothesis of potential cardiovascular risk factors and pathology that may affect females more. Indeed, there is evidence supporting cardiovascular fitness to be protective or vascular risk factors to be harmful towards cognitive decline and dementia in women[42,77–79]. Furthermore, these diagnoses suggest a phenotype for females with AD along with other degenerative diseases of aging and frailty. In particular, the increase in musculoskeletal and bone disorders in females with AD, as well as high calcium and vitamin D deficiency, may point to a potential bone metabolism pathology or aberrant calcium metabolism in females with AD. From a psychiatric standpoint, the female AD phenotype is more associated with depression compared to males as supported by studies that found depression associated with greater hippocampal volume loss in women[80], and is more likely to be a manifestation of mild cognitive impairment or AD in females[81,82].

We performed sensitivity analysis by taking the number of encounters for each group into account. In general, we see a decrease in statistical significance in our enrichment analysis consistently across all diagnoses. This is likely due to decreased power from a lower sample size, and a bias toward the selection of patients with more severe disease due to encounter thresholding.

Overall, enriched diagnoses are relatively similar, with an increase in cerebrovascular disorders observed in AD, and particularly females with AD. Neuroimaging studies have identified differences in AD phenotypes and brain networks depending on the presence of cerebrovascular disease[83,84], which may support cerebrovascular events as an associated phenotype for a different or severe phenotype of AD.

Medication enrichments show expected associations with AD, as the top medication hits are current therapies used to modify symptoms of AD (e.g., memantine, donepezil), or are associated with diagnoses found in comorbidity analysis (e.g., antidepressants for depression).

These medications are also identified as AD-enriched in our validation cohort, although many of these medications are expected as they are associated with conditions of aging. Medications enriched in controls provide a more interesting story, as they not only suggest an 'opposite AD' phenotype, but control-enriched hits may provide a way to hypothesize potential targets for further exploration of protective drug effects or drug repurposing. From our medication analysis, we see control enrichments of opioids, sedatives, dexamethasone, and furosemide, with dexamethasone, also found significant in our validation cohort. The negative association with opioids is inconsistent with prior studies that found associations between prescription opioid use and AD risk[85], although control enrichment of opioids could possibly be due in part to decreased ability to communicate pain and decreased opioid prescriptions after AD[86]. Nevertheless, studies have implicated the role of opioid system dysregulation in tau hyperphosphorylation and AD[87]. Dexamethasone is a corticosteroid that has been suggested to help reduce inflammation in AD[88,89], although the data on efficacy is still uncertain and may depend upon the need for combination therapy[90] or control of other factors that complicate the relationship between hormonal levels and the brain[91,92]. Furosemide is a diuretic drug used to treat hypertension and may confer a protective effect through the control of comorbid conditions that contribute to cardiovascular risk factors. Furosemide also reduces the production of CSF by inhibiting carbonic anhydrase, which may impact CSF dynamics and help decrease the risk of AD[93]. Prior studies have shown possible protective effects from diuretic drugs and AD[94–97], and one study identified furosemide as a potential probe molecule for reducing neuroinflammation[98].

Characterizing patients by lab values provides another way to phenotype patient groups. Through our analysis, greater calcium levels were identified, especially in females with AD. A small observational study found calcium supplementation to increase the risk of dementia in women with cerebrovascular disease[99]. Calcium dysregulation and homeostasis have been implicated in AD neuronal signaling pathology, and identified as a target for drug development[99,100]. Control-enriched labs may also be related to gastrointestinal cancers or liver/pancreatic dysfunction, as we observe increased AST, ALT, and glucose levels in controls and particularly among males. This result is not consistent with a study observing greater glucose levels to increase dementia risk[101], although one study did find low ALT[102] to be associated with AD, and some publications implicate altered glucose metabolism[103,104] and liver dysfunction in AD pathology[102,105,106]. Furthermore, since our control cohort has been matched on age and death status, control patients may encompass a population with a terminal disease. Lab clusters also demonstrate phenotypes specific to a sex group. A lower clotting time (aPTT, PT) and greater platelet count, prealbumin, lymphocytes, and cholesterol levels in females with AD may provide a multivariate way to identify potential AD phenotype in females. Prior studies have shown high thrombin[107,108], abnormalities of hemostasis[109,110], and abnormal platelet activation[111–113] in patients with AD that may contribute

to a pro-thrombotic state in AD[114], leading to microinfarcts and cerebrovascular dysfunction[115,116], although sex-specific associations have not been studied previously. Furthermore, control sex phenotype may demonstrate protective labs or biomarkers that decrease the risk of AD. We see lower free T3 in control males, and greater free T4 in control females. Indeed, studies on AD populations have shown high TSH and low free T4 to be associated with the disease[117–119], although sex-specific associations have not been explored in depth.

Some limitations do exist in our study. First, AD is an insidious and heterogeneous disorder, and is frequently misdiagnosed even in specialized dementia centers. Clinically, Alzheimer's dementia is suspected when disease biomarker status is unknown, whereas Alzheimer's disease is diagnosed when biomarker status is confirmed. Our current study did not rely on biomarker-positive cases of Alzheimer's disease, and we did not exclude patients with other pathologies that can also impact brain health through different pathways, such as Parkinson's disease. Nevertheless, Alzheimer's disease often co-occurs with other dementias[120,121]. Second, EMRs, while a rich data source, is a very sparse data set with a lot of missing data, such as sociological factors (e.g., income, education, etc). Nevertheless, the number of patients represented in the EMR is exceptionally large and provides robust opportunities for deriving meaningful insights or hypotheses. This limitation also applies to our validation EMR. Additionally, some associations may be different across the two systems due to differences in the underlying patient populations or standards of care. Therefore, it is possible that the UCSF EMR does not capture an association that may be more prevalent in a different population in New York, and vice versa. How other covariates including socioeconomic factors modify specific AD associations is a question that can be followed up in future work. Third, our definition of controls comes with limitations, as it is difficult to identify "healthy" controls in the EMR. The institutions represented in our data include both primary and tertiary care, which includes patients that seek hospital care for a variety of reasons. As such, there may be bias in the underlying patient population who chooses to seek medical care at a metropolitan medical center. Regardless, the power in utilizing EMR allows us to generate hypotheses with a large number of patients and versatility in choice of controls compared to many current AD studies. Lastly, our analysis only identifies associations with AD and does not take temporal factors into consideration, therefore causal relationships cannot be concluded. This will be the main focus of future work, as the temporal association can categorize an association as a risk/protective factor (if early in age), a diagnostic clue (if during AD diagnosis), or as a manifestation of AD progression or severity (if after AD diagnosis). Nevertheless, given AD is an insidious disorder, there can be brain perturbations a decade or more before a diagnosis is determined and documented in clinical records. While we made the assumption of independence in our statistical methods to identify significant associations, this method can be further extended to alternative statistical models that take covariates into account. Our current work allows the unbiased identification of associations and phenotyping, which can then be used to generate hypotheses for guiding follow-up studies.

Overall, our analyses leveraged an extensive clinical data set to (1) phenotype and represent AD and (2) perform enrichment analysis to identify known or suggested novel associations with AD, as well as elicit sex-specific differences. We were therefore able to apply an integrative, unbiased, big data approach to identify associations with AD and provide phenotypic representations of an otherwise complex disease. With this approach,

we can generate many new hypotheses to better motivate future work to understand AD complexity and develop diagnostic strategies and therapeutic interventions. Future work will include temporal analysis in order to identify longitudinal relationships and predictive modeling for AD risk, diagnosis, or progression. More extensive analysis of medication and lab values, especially among opposite phenotypes in controls, may lead to better strategies for the prevention or treatment of AD. Besides elucidating sex differences, the next steps for phenotyping can include investigating race/ethnicity differences or differences based upon other covariates to better characterize Alzheimer's Disease heterogeneity. Furthermore, the incorporation of molecular or genetic data with clinical data can help better elucidate potential mechanisms underlying identified associations.

## Methods

All analysis of UCSF and Mount Sinai EMR data was performed under the approval of respective Institutional Review Boards. All clinical data were de-identified and written informed consent was waived by the institutions.

In this study, we performed deep phenotyping and association analysis of patients with AD and controls. First, AD and control cohorts were identified from the UCSF EMR and topographically visualized via a low-dimensional projection of comorbidities. Comorbidity networks were created, and association and enrichment analyses were performed on all diagnoses, medications, and lab values. These analyses were further performed in a sex-stratified manner to identify sex-specific associations, and validation was performed on the Mount Sinai EMR. An overview of the workflow is shown in Fig. 1.

**Patient cohort identification.** Patient cohorts were identified from over five million patients in the UCSF EMR database, which includes clinical data from 1982 to 2020. Due to the de-identification process, dates are shifted by at most a year (with relative dates preserved) and all birth dates before 1930 (=estimated age 90) are shifted to be no earlier than 1930. Patients with AD were identified by inclusion criteria of estimated age >64 years, and ICD-10-CM codes G30.1, G30.8, or G30.9, where estimated age is determined from the birth date. Male and female groups were identified by the most recent sex assignment in the EMR. To identify a control group, we used propensity score (PS) matching method (matchit R package[115]) by a logistic regression model to match controls to patients with AD. The control group was selected from patients >64 years old without AD diagnosis, matched on sex, estimated age, race, and death status at a 1:2 AD:control ratio using a nearest neighbors method. The validation cohort was identified similarly in the Mount Sinai EMR database, which includes clinical data from 2003 to 2020. The demographic properties of the UCSF and Mount Sinai cohorts are shown in Table 1.

**Dimensionality reduction patient visualization.** All identified patients were represented with one-hot encoding of diagnoses, excluding encoding of diagnoses with Alzheimer's in the name (list in Supplementary Table 2 and Fig. 2). Patients were then visualized in a lower dimension using Uniform Manifold Approximation and Projection[122] (UMAP) with the umap-learn package from Python. Correlations between variables and UMAP coordinates were analyzed using Mann–Whitney U-test for categorical variables, and Pearson's correlation coefficient for continuous variables.

**AD vs. control enrichment analysis of comorbidities.** To evaluate comorbidities, all diagnoses recorded from patient cohorts were identified with the earliest entry of every diagnosis. Comparisons were made at different ICD-10-CM hierarchical levels, specifically Level 2 categories (e.g., G30-G32: Other degenerative diseases of the nervous system), Level 3 categories (e.g., G30: Alzheimer's Disease), or full diagnosis names (e.g., G30.9 Alzheimer's disease, unspecified). Level 2, Level 3, and full diagnosis names are also grouped by ICD-10-CM blocks (e.g., G00-G99: Diseases of the Nervous System). More information on ICD-10-CM codes can be found here: https://www.cms.gov/Medicare/Coding/ICD10/ICD-10Resources.

Diagnosis networks were created based upon a diagnosis category or diagnosis shared by >1% patients in a group (node) or pair of diagnosis categories or diagnoses shared by >1% of patients in a group (edge). Network metrics were computed using Cytoscape app Network Analyzer[123]. Metrics were then compared between AD and control networks using Mann–Whitney U-test, with and without singleton nodes removed. Nodes and edges were thresholded by 5% of patients in a group for visualization purposes.

Enrichment analysis of diagnosis was compared between AD and control cohorts. For each diagnosis, the proportions of patients in each group were compared using Fisher's exact (if <5 patients in a category) or Chi-squared test. Significant diagnoses were determined by a Bonferroni-corrected threshold of p-value < 0.05, and directionality determined with odds ratio (OR). With

inspiration from genetic and molecular approaches, the results were visualized using Manhattan plots by categorizing diagnoses in ICD-10-CM blocks.

**Sex-stratified AD vs. control enrichment analysis of comorbidities**. Diagnostic networks were created for each sex, with diagnosis categories or diagnoses shared by >1% of patients in a group (node), and diagnosis category/diagnosis pair shared by >1% of patients in a group (edge). Network metrics were then computed using Cytoscape Network Analyzer app, and compared between sex-stratified patients with AD and controls, and between males and females for both AD and control cohorts separately with a Mann–Whitney *U*-test. Nodes and edges were thresholded by 5% of patients in a group for visualization.

Sex-specific enrichment analysis of diagnoses between AD and control cohorts were compared with a subset of equal numbers of patients with AD and controls within each sex. For each diagnosis, the proportions of patients in each group were compared using the Fisher's exact (if <5 patients in a category) or Chi-squared test. Significance was determined by applying a threshold of 0.05 for Bonferroni-corrected *p*-values. Log–log plots were generated from odds ratios between females and males with AD and controls, and Miami plots were created by categorizing diagnoses in ICD-10-CM blocks.

**Sensitivity analysis taking encounters into account**. Sensitivity analysis of diagnosis enrichment analysis was performed with a subgroup of patients with AD and a second control cohort to account for variability in the number of visits for each patient. AD cohorts were subgrouped by identifying patients with over 10 encounters in the EMR and records spanning over a year. The encounter-filtered control cohort was identified by additionally matching the number of encounters and years between the first and last record in the EMR. Diagnosis enrichment analysis was carried out as described above for general comorbidities and sex-specific analysis.

**AD vs. control enrichment analysis of medications**. All medications ordered for patients with AD and controls were extracted and grouped based upon the generic medication name, with route and dosage information removed. The proportions of patients with AD and controls prescribed each medication were compared using Fisher's exact (if <5 patients in a category) or Chi-squared tests. Significantly enriched medications were identified by a Bonferroni-corrected threshold of *p*-value 0.05, and directionality was determined with an odds ratio. Sex-specific medication comparisons were also performed within a subset of equal numbers of patients with AD and controls for each sex and plotted with cutoffs based upon a Bonferroni-corrected *p*-value threshold of 0.05 and odds ratios threshold of <0.5 or >2.

**AD vs. control comparisons of lab values**. For laboratory values, median values for all numerical lab test results for each patient were identified. Lab tests missing data among 95% or more patients were removed. Lab value distributions were compared using Mann–Whitney *U*-test across three comparisons (AD vs. controls, females with AD vs. female controls, and males with AD vs. male controls) in order to identify significantly different lab values.

For clustering analysis, significant lab tests above a threshold of 0.05 for Bonferroni-corrected *p*-value were isolated, and mean values were then identified for each group (females with AD, males with AD, control females, control males) and normalized across groups as a *Z*-score. Clustering was then performed using the sigclust2 R package[124] to determine the significance of each cluster break using permutations (Euclidean distance metric and average linkage).

**Validation in external EMR**. AD and PS-matched control patients were identified in the Mount Sinai EMR in the same fashion as described in [Patient Identification] in the UCSF EMR. All aforementioned analyses with dimensionality reduction, comorbidity networks, diagnosis/medication enrichments, sex-specific enrichments, and lab value comparisons were performed in the Mount Sinai data set as they have been in the UCSF EMR data set.

For network comparisons, network metrics were standard normalized across the 12 networks (6 at UCSF, 6 at Mount Sinai) by the metric and Spearman-rank correlation coefficient and significance determined. For diagnosis comparison, Level 2, Level 3, and full diagnosis names were mapped and compared by the sub-chapter, three-digit codes, and full code of the ICD-10-CM hierarchy, respectively. Significant diagnosis in the validation cohort was determined by a Bonferroni-corrected threshold of 0.05 based upon the number of mapped UCSF-significant diagnoses. Correlations between odds ratios were determined by a Spearman-rank correlation coefficient and significance. Medications were mapped based upon the generic name, and correlations between odds ratios determined with the Spearman-rank correlation coefficient.

For comparison of labs, the normalized lab values for each institution were combined, and clustering was performed using Euclidean distance and average linkage to identify groups of labs with similar trends between AD/sex/institution stratified patient groups. The R package *sigclust2* was used to determine significant clusters of labs.

**Data visualization using RShiny**. An interactive visualization of comorbidity enrichments and networks between AD and control groups and with sex stratification was implemented in an Rshiny[125] app: vizad.org.

**Reporting summary**. Further information on research design is available in the Nature Research Reporting Summary linked to this article.

## Data availability

The UCSF EHR database is available to individuals affiliated with UCSF who can contact the UCSF's Clinical and Translational Science Institute (CTSI) (ctsi@ucsf.edu) or the UCSF's Information Commons team for more information (Info.Commons@ucsf.edu). The Mount Sinai EHR database is available to individuals affiliated with Mount Sinai who can contact the Mount Sinai Intellectual Partners (MSIP) for more information (MSIPInfo@mssm.edu). If the reader is not affiliated with the aforementioned institutions, they can set up an official collaboration with an investigator affiliated with the target institution(s) by contacting the PIs Marina Sirota (marina.sirota@ucsf.edu) and Benjamin Glicksberg (benjamin.glicksberg@mssm.edu). Requests should be processed within a couple of weeks. Summary data is available in supplementary files and, for UCSF, can be explored at https://vizad.org.

## Code availability

The code is available at https://github.com/al1563/adehr_phenotype.

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

## Acknowledgements

Primary support through Grant # NIA R01AG060393, R01AG057683 (A.T., T.O., C.W.S., M.S.). Additional support was provided by NIA RF1AG068325 (D.B.D) and Medical Scientist Training Program T32GM007618 (A.T.). B.G. and M.B. are supported by grant 1 RF1 AG059319-01. We'd like to acknowledge Zachary Cutts, Stella Belonwu, and other members of the Sirota Lab for their suggestions and help.

## Author contributions

A.T. and M.S. designed the question, experiments, and analytic plan. B.Z. and Z.H. helped with data acquisition, cleaning, and interpretation. A.T., C.W.S and B.O. helped with creation of Rshiny app. A.T., W.M., T.O., C.W.S and D.D. interpreted results. J.H., M.B. and B.G. helped acquire and analyze validation data. S.W. and I.A. aided in statistical methods. A.T. wrote the manuscript with editing from all the authors. All the authors edited and reviewed the manuscript.

## Competing interests

The authors declare no competing interests.
