## [Peer review file · Nature Communications]

Reviewers' comments:

Reviewer #1 (Remarks to the Author):

This paper investigates the clinical factors associated with AD risk using electronic medical records with a focus on sex specific associations.

There are two conceptual issues in the paper : one is about the use of Alzheimer's disease. The authors are in fact investigated AD dementia (and I will challenge the definition after) and this should be made clearer. Second I tend to disagree with the authors statements that women at higher risk of AD dementia as established. I think there is evidence that this could be fully related to selection and one should rather correct for this bias rather than trying to explain biased associations.

Diagnosis of dementia is complex and age at diagnosis is crucial to establish risk factors. The use of EMR is challenging and it would be crucial to undertake validity analyses. We need some data to demonstrate the validity of the outcome measurement

In the results, it is quite surprising to see vascular dementia as a top diagnoses in AD. This rather looks like misclassification

The use of EMR is interesting because it allows an agnostic search for risk factors but it also has some important pitfalls including some potential confounding factors that are not taken into account especially in this case education or social factor. This needs to be discussed

The authors performed stratified diagnoses by sex but how can they prove that the gender differences are meaningfully different

Given the data weaknesses, this work would be reinforced if replicated in a different EMR database

Reviewer #2 (Remarks to the Author):

This manuscript utilized EMRs and performed network analysis to provide insight into clinical characteristics associated with sex.

Their network showed patients with AD

have greater comorbidity interactions in AD. Medication and lab result associations also showed similar results. This manuscript has several major concern.

1. The authors have performed network analysis and presented in different ways. However, why the authors are mentioning deep clinical phenotyping?

2. The authors used existing methodology and there is no method section to the manuscript. How they have calculated the association/correlation between the two diseases and as well as with the medications? There is some methodology for the comorbidity risk calculation, such as phi-correlation, relative risk.

3. Sometimes the network analysis gives random result. So it is better to validate using some statistics approach such as hypergeometric test or using different datasets, such as WGS, GWAS, WES data, can be used to support the result. Otherwise, it is also possible to use different cohort of data.

4. Is there any difference between ethnicity? It could be other findings.

5. Figure 1 and Figure 2 are not giving any important results.

6. Caption of the figures should be meaningful, it should provide some information that you observed in this analysis (main findings). It is mentioned this figure is fr this.... but need to provide what did you found.

7. In this study it is important to identify the causal relationship/ causal inference between the

disease. It is possible by considering the occurrence time of disease for each patient.

8. Result section needs to be modified. Should be written mostly what did you find in your analysis rather than which type of representation.

Reviewer #3 (Remarks to the Author):

This paper aimed to investigate an important question about the phenotypic heterogeneity for Alzheimer's Disease. The authors sampled a nested case control study from a large scale electronic medical record database, using propensity score matching with 1:2 case-control sampling ratio. After frequency matching with age, sex, race, and death status, the authors performed a series of extensive descriptive analyses, including UMAP, comorbidity network analysis, bi-clustering, and diagnostic-wise odds ratio across interested groups (patient vs controls, male vs female, etc). The authors concluded patients with AD have different clinical profile than controls, in terms of comorbidities, lab values, and medication received, while sex differences were also observed. While the topic is indeed extremely important and I found the results most interesting in comparing comorbidities between sexes, there are some major issues that need to be addressed.

Major:

1. The conflicting goals between case-control selection and unsupervised learning: Because the study is hospital based (EMR), the controls were essentially defined as "those who were seeking hospital treatment other than Alzheimer's Disease". The impact became more complicated when the matching is introduced, as the controls were further selected as older individuals who were seeking hospital treatment. Therefore, by design, the diagnostic profiles would, by default, be different between case and controls, which is not necessarily reflecting the properties of AD cases but the special conditions of controls. The enrichment of malignant cancers among controls from the authors' results is an example of such issue. By imposing unsupervised learning, such as UMAP and bi-clustering, on top of the selected samples, it beats the purpose of case-control selection and ignore the fact the control status is completely explained away by the definition of cases.

One way the UMAP can be helpful in current context, is to perform UMAP on the cohort level before the selection (UCSF EMR dataset), using the features that enhance the validity of the afterwards comparisons, to demonstrate a). the properties of the cohort that sampling based upon, b). the selection distribution when the criteria is imposed, and c). the appropriateness of the propensity score construction. Nevertheless, when the number of features is limited, as the confounding variables listed in current paper, a table that summarize a). entire cohort, b). cases, and c). sampled cohorts might be sufficient without the need for using the unsupervised methods.

2. The issue of statistical method for controlling confounds: The strength of nested case control study through matching design is to facilitate the efficient analyses while controlling for potential confounds. Because the selection is introduced by design, any analysis does not take into consideration of the selection would introduce the collider bias, making unrelated variables related (Austin, PC, 2008, *Statistics in Medicine*, 2008, 27:2037-2049). In current paper, all the group comparisons, including the construction of networks and enrichment analyses, were not taking into account of the introduced selection. The propensity score matching is the first step to enabling the confounding control, not the completion of the confounding controlling by itself, therefore, the conclusion based on current results can be fairly limited. Appropriate methods should be used when the matching is imposed here, such as conditional logistic regression or GEE (Austin, PC, 2008, *Statistics in Medicine*, 2008, 27:2037-2049).

Minor:

1. The matching score should be called disease risk score rather than propensity score, because the matching scheme is used for matching case/controls, not the exposure/treatment status (Rothman, Modern Epidemiology; Desai et al., 2016, American Journal of Epidemiology, 10, 949-957).
2. Unclear what is the age used for the matching scheme. Is it age at diagnosis or age of first EMR entry?
3. Table 1. has mis-specified the proportion of males and females among AD.
4. Need to define selection process more clearly, such as a). which years were considered eligible EMR entries, b). demographic properties of the main EMR cohorts, c). the sampling proportions based on the imposed criteria, d). the exclusion criteria imposed.
5. It would be good to perform sensitivity analyses to include the temporal ordering of the diagnoses. For example, restricting the comorbidities ascertained only before the AD diagnoses were made while the controls matched with the event occurred as the risk set.

Although we cannot offer to publish your paper in Nature Communications, the work may be appropriate for another journal in the Nature Research portfolio. If you wish to explore suitable journals and transfer your manuscript to a journal of your choice, please use our <https://mts-ncomms.nature.com/cgi-bin/main.plex?el=A2S3CJny3A6FPEI4X7A9ftdX4rIaisMwN9yYVKZbghhAZ> manuscript transfer portal. If you transfer to Nature-branded journals or to the Communications journals, you will not have to re-supply manuscript metadata and files. This link can only be used once and remains active until used.

All Nature Research journals are editorially independent, and the decision to consider your manuscript will be taken by their own editorial staff. For more information, please see our http://www.nature.com/authors/author_resources/transfer_manuscripts.html?WT.mc_id=EMI_NPG_1511_AUTHORTRANSF&WT.ec_id=AUTHOR manuscript transfer FAQ page. Note that any decision to opt in to In Review at the original journal is not sent to the receiving journal on transfer. You can opt in to *[In Review](https://www.nature.com/nature-research/for-authors/in-review)* at receiving journals that support this service by choosing to modify your manuscript on transfer. In Review is available for primary research manuscript types only.

Reviewer Comments:

We would like to thank the Reviewers and Editors for their helpful comments and comprehensive review of our work titled “Deep Clinical Phenotyping of Alzheimer’s Disease Patients Leveraging Electronic Medical Records Data Identifies Sex-Specific Clinical Associations”. Please find our point by point responses below in blue.

Reviewer #1 (Remarks to the Author):

This paper investigates the clinical factors associated with AD risk using electronic medical records with a focus on sex specific associations.

There are two conceptual issues in the paper : one is about the use of Alzheimer's disease. The authors are in fact investigated AD dementia (and I will challenge the definition after) and this should be made clearer. Second I tend to disagree with the authors statements that women at higher risk of AD dementia as established. I think there is evidence that this could be fully related to selection and one should rather correct for this bias rather than trying to explain biased associations.

We would like to thank the reviewer for bringing up these issues and suggestions. We have clarified the discrepancy in the usage of the terminology of Alzheimer’s Disease and Alzheimer’s dementia in our limitations. Within our EMR, “Alzheimer’s Disease” was the diagnosis corresponding to the ICD-10 codes we examined. At least 1/3 of our patients visit the Memory and Aging Center at UCSF, where there are biomarker studies and trained neurologists whose expertise allow us to increase our certainty of the diagnosis of Alzheimer’s Disease. In terms of the sex difference issue, we have clarified the sex difference risk in the paper. Prevalence studies have shown that more women than men have Alzheimer’s or other dementias, and almost 2/3 of Americans with Alzheimer’s are women (**Rajan, K. B., Weuve, J., Barnes, L. L., et al. *Alzheimer's & Dementia*, 2021;17**), and women also have increased estimated lifetime risk for Alzheimer’s dementia (**Chene G, Beiser A, Au R, et al. *Alzheimers Dement* 2015;11(3):310-320.**). Nevertheless, there is mixed evidence of the risk of developing Alzheimer’s dementia between men and women of the same age, which the reviewer may be alluding to (**Neu SC, Pa J, Kukull W, et al. *JAMA Neurol.* 2017;74(10):1178–1189, Katrine L. Rasmussen, et al. *CMAJ Sep 2018, 190 (35) E1033-E1041, Matthews, F., et al. *Nat Commun* 7, 11398 (2016)***). We have clarified this in the introduction on page 2 and 3. Nevertheless, we provide references to multiple studies to demonstrate that sex contributes to overall differences in AD risk, manifestation, pathology, and biology, which all show that sex is an important covariate to consider in studying Alzheimer’s Dementia, and we aim to contribute to this understanding through our paper leveraging EMR datasets.

Diagnosis of dementia is complex and age at diagnosis is crucial to establish risk factors. The use of EMR is challenging and it would be crucial to undertake validity analyses. We need some data to demonstrate the validity of the outcome measurement

We would like to thank the reviewer for their comments and insight. We agree that electronic medical records are a challenging and heterogeneous dataset to work with which is mentioned in the limitations portion of the discussion on page 17, but we believe it is also a vast resource that can be leveraged to derive insight into diseases. We have taken the reviewer's comment into consideration and have since repeated our analysis in an external independent electronic medical record at Mount Sinai. In general, we observe strong and significant correlations between the findings between the two EMRs. These findings are now described in the paper throughout the results section and included in Figures 1-6 and Supplementary Figures 1-4. We recognize that the external EMR may also be limited in its ability to capture the patient population in our original dataset, due to differences in the underlying location and standards of clinical care -- this is a limitation we mention in page 17.

In the results, it is quite surprising to see vascular dementia as a top diagnoses in AD. This rather looks like misclassification

We would like to thank the reviewer for bringing up this point. Our selection of Alzheimer's dementia patients is broad, which means we may include patients with mixed dementias or potential misclassification. In one study, over 50% of Alzheimer's patients that met pathologic criteria show evidence of coexisting dementias (**Brenowitz, Willa D., et al. Alzheimer's & Dementia 13.6 (2017):654-662**), and other studies suggest mixed dementia is the norm and that cerebrovascular disease manifests more commonly as mixed pathology, (**2021 Alzheimer's disease facts and figures. Alzheimers Dement., 17: 327-406**), with AD and vascular dementia being a common combination. Another EMR study identified 7.5% overlap between AD and vascular dementia diagnosis, which may be due to either mixed dementia or misclassification (**Jørgensen, IF, Aguayo-Orozco, A, Lademann, M, Brunak, S. Alzheimer's Dement. 2020; 16: 908– 917.**). In clinical practice, investigating amyloid/tau pathology is not necessarily possible before death. Nevertheless, we believe there is still value in studying this cohort of Alzheimer's dementia, as the disease may be more practically characterized through a clinical lens. We have added this caveat to the limitation section of the discussion on page 18.

The use of EMR is interesting because it allows an agnostic search for risk factors but it also has some important pitfalls including some potential confounding factors that are not taken into account especially in this case education or social factor. This needs to be discussed
The authors performed stratified diagnoses by sex but how can they prove that the gender differences are meaningfully different
Given the data weaknesses, this work would be reinforced if replicated in a different EMR database

We would like to thank the reviewer for first recognizing the value of our work but also bringing up these limitations. We recognize the heterogeneity of EMR data which is now stressed in the discussion section on page 18. In order to avoid potential confounders in our study, we matched our controls on covariate factors that are commonly considered confounders in literature. While the confounders you mentioned, such as education or social factors, are important, these

variables are not readily available in the EMR. We have explicitly added this limitation in our paper on page 18.

In terms of replication, we have repeated our analysis in an external independent dataset (Mount Sinai), where patients may come from different background characteristics. In general, we find significant correlations between the association results of the two independent analyses in diagnosis, medications, and labs across UCSF and Mount Sinai. The findings and some specifics are now included in Figure 1-6 and Supplementary Figures 1-4. While we agree that EMR data is highly heterogeneous and complicated, we believe there is still insight in the EMR that can be useful to extract, particularly in terms of power, to complement and help progress AD research.

Reviewer #2 (Remarks to the Author):

This manuscript utilized EMRs and performed network analysis to provide insight into clinical characteristics associated with sex.

Their network showed patients with AD have greater comorbidity interactions in AD. Medication and lab result associations also showed similar results. This manuscript has several major concern.

We would like to thank the reviewer for summarizing our work. We address the concerns point by point below.

1. The authors have performed network analysis and presented in different ways. However, why the authors are mentioning deep clinical phenotyping?

We would like to thank the reviewer for reviewing our paper and bringing up good questions. In our study, we utilized electronic medical record data, which provides comprehensive and extensive data, to investigate our patients. In general, the term 'deep phenotyping' has been used in precision medicine approaches to provide more detailed stratification and representation of a disease (**Delude, C. Nature 527, S14–S15 (2015); Weng C, Shah NH, Hripcsak G. J Biomed Inform. 2020;105:103433.**). In our case, providing a representation of our cohort using diagnostic, medication, and lab data (and via networks) allows us to better characterize and perform an extensive association analysis with our cohort. We clarified this on page 4 and 5.

2. The authors used existing methodology and there is no method section to the manuscript. How they have calculated the association/correlation between the two diseases and as well as with the medications? There is some methodology for the comorbidity risk calculation, such as phi-correlation, relative risk.

Thank you, we apologize if anything is unclear with regards to the methods. There is a separate document for Online Methods that provides more details on the methodology, including the statistical method used for comparing cases and controls using fisher exact or chi-square test

(AD vs. Control Enrichment Analysis of Comorbidities of Methods, page 21). We have now brought the methods section back into the main text of the manuscript on page 20 (shown in blue in the tracked document).

3. Sometimes the network analysis gives random result. So it is better to validate using some statistics approach such as hypergeometric test or using different datasets, such as WGS, GWAS, WES data, can be used to support the result. Otherwise, it is also possible to use different cohort of data.

Thank you for your suggestions. We apologize if anything is unclear with regards to the methods. We are applying robust statistical approaches to compare network metrics across groups (AD vs. Control Enrichment Analysis of Comorbidities in the Methods, page 21) to validate our network analysis. Moreover, we have also included an independent validation of the network analysis on the Mount Sinai EMR which is now shown in Figure 3. Thank you for your suggestion regarding WGS, WES and GWAS. While this is interesting data to explore, it is outside of the scope of this work since we focus on clinical data here and we do not have molecular nor genetic data on these patients. We added integrating molecular and genetic data to the future directions on page 19.

4. Is there any difference between ethnicity? It could be other findings.

We would like to thank the reviewer for bringing up this important question. We have begun investigating race/ethnicity differences in AD clinical associations using EMR data. We found some interesting preliminary results that seem consistent with and potentially expand on prior studies that have looked at these differences (**Ladecola C, Yaffe K, Biller J, et al. Hypertension. 2016;68(6):e67-e94., Levine DA, Gross AL, Briceño EM, et al. JAMA Neurol. 2020; 77(7):810-819.**). Socioeconomic status, racism, and other sociological factors may contribute to the differences that we observe (**Yaffe, Kristine, et al. Bmj 347 (2013).**). Additionally, since genetic ancestry can be heterogeneous within racial categories, this could likely complicate biological interpretations of these results (**Belbin GM, Cullina S, Wenric S, et al. Cell. 2021 Apr 15;184(8):2068-2083.e11.**). Therefore, we think that exploring race/ethnicity differences, which we hypothesize to be likely due, in large part, to differences in sociological (not necessarily biological) factors, is beyond the scope of this paper and is a topic of a follow up manuscript that we are currently working on. Since this is an important question, we have added this analysis as a future direction on page 18.

5. Figure 1 and Figure 2 are not giving any important results.

Thank you for your comment. Figures 1 and 2 are meant to give an overview of the study design and data to provide readers a high-level overview of the data and approaches we use to accomplish our research goals. Therefore, we would like to keep these in the main text.

6. Caption of the figures should be meaningful, it should provide some information that you observed in this analysis (main findings). It is mentioned this figure is for this.... but need to provide what did you found.

Thank you for your suggestion. The captions of the figures have been updated to provide more information.

7. In this study it is important to identify the causal relationship/ causal inference between the disease. It is possible by considering the occurrence time of disease for each patient.

Thank you for your comment. This is a great suggestion and one we hope to implement in the future. In this paper, we did not look at temporal or causal relationships, as this will be a topic of focus in future analysis. We mention this as a limitation and a future direction on Page 18.

8. Result section needs to be modified. Should be written mostly what did you find in your analysis rather than which type of representation.

Thank you for your suggestion. We have updated the Results section to focus on the findings. Specifically, we edited the subheadings to reflect the findings. For instance, "**Alzheimer vs. Control Association Analysis Identifies Previously Known and Novel Associated Comorbidities in AD**" instead of "Case Control Enrichment Analysis of Comorbidities".

Reviewer #3 (Remarks to the Author):

This paper aimed to investigate an important question about the phenotypic heterogeneity for Alzheimer's Disease. The authors sampled a nested case control study from a large scale electronic medical record database, using propensity score matching with 1:2 case-control sampling ratio. After frequency matching with age, sex, race, and death status, the authors performed a series of extensive descriptive analyses, including UMAP, comorbidity network analysis, bi-clustering, and diagnostic-wise odds ratio across interested groups (patient vs controls, male vs female, etc). The authors concluded patients with AD have different clinical profile than controls, in terms of comorbidities, lab values, and medication received, while sex differences were also observed. While the topic is indeed extremely important and I found the results most interesting in comparing comorbidities between sexes, there are some major issues that need to be addressed.

We would like to thank the reviewer for summarizing our work and recognizing the value of this study. We address the concerns point by point below.

Major:

1. The conflicting goals between case-control selection and unsupervised learning: Because the study is hospital based (EMR), the controls were essentially defined as "those who were seeking hospital treatment other than Alzheimer's Disease". The impact became more complicated when the matching is introduced, as the controls were further selected as older

individuals who were seeking hospital treatment. Therefore, by design, the diagnostic profiles would, by default, be different between case and controls, which is not necessarily reflecting the properties of AD cases but the special conditions of controls. The enrichment of malignant cancers among controls from the authors' results is an example of such issue. By imposing unsupervised learning, such as UMAP and bi-clustering, on top of the selected samples, it beats the purpose of case-control selection and ignore the fact the control status is completely explained away by the definition of cases.

One way the UMAP can be helpful in current context, is to perform UMAP on the cohort level before the selection (UCSF EMR dataset), using the features that enhance the validity of the afterwards comparisons, to demonstrate a). the properties of the cohort that sampling based upon, b). the selection distribution when the criteria is imposed, and c). the appropriateness of the propensity score construction. Nevertheless, when the number of features is limited, as the confounding variables listed in current paper, a table that summarize a). entire cohort, b). cases, and c). sampled cohorts might be sufficient without the need for using the unsupervised methods.

We would like to thank the reviewer for insightful comments and suggestions. We apologize for any confusion on the goals of our analysis. While your suggestion is great for utilizing UMAP to identify features that may separate cases and controls, because our patient population is so large, it is not computationally feasible to visualize the whole UCSF cohort (> 5 million patients) using a UMAP. Our approach was to use unsupervised approaches simply as a sanity check to visualize separation between cases and controls from the features (diagnosis only) that we choose with low-dimensional embedding. In other words, we are asking if we can observe separation by representing cases and controls by diagnosis information only, and performing UMAP acts as a means of validating that to be the case. We have clarified this on page 4 of the manuscript and also have demographic information in Table 1 as suggested by the reviewer.

In terms of defining controls, any definition of controls will have limitations. UCSF is a clinical center that includes both primary care and tertiary care, and the patient population involves a wide net of patients that seek hospital care for a variety of reasons, from routine visits to complicated clinical cases. All of these visits are recorded in the EMR, not just hospital-based visits. According to the CDC, in 2019, 84.9% of adults in the U.S. have seen a healthcare professional, so we can expect the background population to be a sample of that.

Furthermore, it is difficult to define a 'healthy' control. The covariates that we control for are determined a priori to account for criteria that we determine are important to control for, so that diagnostic results are not likely to be attributed to those covariates. All choices of control cohorts will have their limitations, and we acknowledge that our controls will have limitations as well. We have added several sentences to the discussion section on page 18 stating the aforementioned limitations. Nevertheless, the power of utilizing EMR allows us a greater number of patients and versatility in our choice of controls compared to many current AD studies. We have replicated our analysis in an independent EMR (Mount Sinai), which we have included in our paper and

figures. Finally permutation analysis as well as validation in an independent EMR strengthen our findings.

2. The issue of statistical method for controlling confounds: The strength of nested case control study through matching design is to facilitate the efficient analyses while controlling for potential confounds. Because the selection is introduced by design, any analysis does not take into consideration of the selection would introduce the **collider bias**, making unrelated variables related (Austin, PC, 2008, *Statistics in Medicine*, 2008, 27:2037-2049). In current paper, all the group comparisons, including the construction of networks and enrichment analyses, were not taking into account of the introduced selection. The propensity score matching is the first step to enabling the confounding control, not the completion of the confounding controlling by itself, therefore, the conclusion based on current results can be fairly limited. Appropriate methods should be used when the matching is imposed here, such as conditional logistic regression or GEE (Austin, PC, 2008, *Statistics in Medicine*, 2008, 27:2037-2049).

We would like to thank the reviewer for the insightful suggestions and comments. The covariates we choose for our matching include common variables that are considered in literature. We would like to thank the reviewer for sharing the useful references with us. In terms of the cited paper, we assessed the balance of covariates between AD and control subjects shown in Table 1. We agree that matching may cause our AD group and controls to no longer be independent samples, but we made the assumption of independence in order to perform Fisher exact or chi-square tests. We acknowledge this as a limitation of our work and indicate that in our discussion on page 18. With regards to concern of collider bias, in one study it was seen from simulations that resulting collider bias is of a minor concern in real world scenarios (Liu, Wei, Alan Brookhart, and Soko Setoguchi. *Pharmacoepidemiology and Drug Safety* 19 (2010)), and even if there was some bias, the reduction in bias from adjustment outweighs any increase in bias due to colliders (Greenland, Sander. *Epidemiology: May 2003 - Volume 14 - Issue 3 - p 300-306*).

Nevertheless, we performed conditional logistic regressions, conditioning on single comorbidities and examining the coefficients and odds ratios to ensure that there was minimal collider bias (e.g. given the existence of a comorbidity, we applied the model $isAD \sim Age + Sex + Race + DeathStatus$). Below are examples of comorbidity results in logistic regression predicting AD vs. Control group. These models have very small coefficients and show no clinically important differences between the outcome groups. These are statistically significant simply because of the large sample sizes. These show that there was no subgroup imbalance creating collider bias and that the propensity scoring was sufficient in the comorbidity analyses.

- Disorders of lipoprotein metabolism and other lipidemias
Logit Regression Results

```
=====
Dep. Variable:          isAD    No. Observations:          6435
Model:                  Logit    Df Residuals:              6431
Method:                  MLE     Df Model:                  3
Date:                   Mon, 19 Jul 2021    Pseudo R-squ.:            0.008894
Time:                   15:31:13    Log-Likelihood:           -4417.0
```

```

converged: True LL-Null: -4456.6
Covariance Type: nonrobust LLR p-value: 4.396e-17
=====
              coef   std err          z      P>|z|     [0.025   0.975]
-----+-----
age             0.0012    0.001     1.169    0.242    -0.001    0.003
Sex            -0.0692    0.050    -1.376    0.169    -0.168    0.029
Race           -0.1351    0.027    -5.007    0.000    -0.188   -0.082
DeathStatus     0.3834    0.052     7.434    0.000     0.282    0.485
=====
              5%      95% Odds Ratio
age           0.999214  1.003118  1.001164
Sex           0.845454  1.029846  0.933106
Race         0.828685  0.921101  0.873672
DeathStatus  1.326228  1.623410  1.467314

```

- Essential (primary) hypertension

Logit Regression Results

```

=====
Dep. Variable: isAD No. Observations: 9874
Model: Logit Df Residuals: 9870
Method: MLE Df Model: 3
Date: Mon, 19 Jul 2021 Pseudo R-squ.: 0.007786
Time: 15:31:13 Log-Likelihood: -6771.5
converged: True LL-Null: -6824.7
Covariance Type: nonrobust LLR p-value: 6.961e-23
=====
              coef   std err          z      P>|z|     [0.025   0.975]
-----+-----
age           -0.0023    0.001    -2.831    0.005    -0.004   -0.001
Sex           -0.0507    0.041    -1.228    0.220    -0.132    0.030
Race          -0.0839    0.020    -4.105    0.000    -0.124   -0.044
DeathStatus   0.4042    0.042     9.636    0.000     0.322    0.486
=====
              5%      95% Odds Ratio
age           0.996199  0.999308  0.997752
Sex           0.876557  1.030735  0.950525
Race         0.883388  0.957089  0.919501
DeathStatus  1.379852  1.626439  1.498081

```

- Other disorders of urinary system

Logit Regression Results

```

=====
Dep. Variable: isAD No. Observations: 4455
Model: Logit Df Residuals: 4451
Method: MLE Df Model: 3
Date: Mon, 19 Jul 2021 Pseudo R-squ.: 0.01347
Time: 15:31:13 Log-Likelihood: -2968.0
converged: True LL-Null: -3008.5
Covariance Type: nonrobust LLR p-value: 1.805e-17
=====
              coef   std err          z      P>|z|     [0.025   0.975]
-----+-----
age             0.0010    0.001     0.826    0.409    -0.001    0.003
Sex             0.1062    0.070     1.506    0.132    -0.032    0.244
Race           -0.0745    0.030    -2.449    0.014    -0.134   -0.015
DeathStatus     0.5105    0.061     8.360    0.000     0.391    0.630
=====

```

	5%	95%	Odds Ratio
age	0.998647	1.003332	1.000987
Sex	0.968509	1.276788	1.112016
Race	0.874440	0.985215	0.928176
DeathStatus	1.478175	1.877923	1.666103

Minor:

1. The matching score should be called disease risk score rather than propensity score, because the matching scheme is used for matching case/controls, not the exposure/treatment status (Rothman, Modern Epidemiology; Desai et al., 2016, American Journal of Epidemiology, 10, 949-957).

We would like to thank the reviewer for this comment. We acknowledge that propensity score has been used to match treatment groups to investigate treatment outcomes and might lead to confusion. We have clarified on page 20 that in our case, it is used to match cases and controls. From our literature search, 'propensity score' has been used previously as the terminology for matching cases and control for looking at association (e.g. **Sacco P, Unick GJ, Zanjani F, Camlin EA. J Dual Diagn. 2015;11(1):83-92.** , **Baek S, Park SH, Won E, Park YR, Kim HJ. Korean J Radiol. 2015;16(2):286-296.** , **Qin Y, Zhang S, Shen X, et al. Therapeutic Advances in Endocrinology and Metabolism. Jan 2019., Ma M, Zhu M, Zhuo B, Li L, Chen H, et al. J Clin Lab Anal. 2020 Aug;34(8):e23313., Kaya A, Isik T, Kaya Y, Enginyurt O, et al. Clin Appl Thromb Hemost. 2015 Mar;21(2):160-5.**). Disease risk score is also used to refer to future risk of a disease, while we are just looking at association and making no temporal statements. In our future work, we plan to look at disease risk scores via building predictive models and don't want to cause any confusion in the terminology that is used.

2. Unclear what is the age used for the matching scheme. Is it age at diagnosis or age of first EMR entry?

We would like to thank the reviewer for bringing this up and we apologize if there was any confusion with the definition of age. The age in the matching scheme is determined by the time of analysis (based upon the birthdate of the patient). We have updated the paper to clarify this point on page 20.

3. Table 1. has mis-specified the proportion of males and females among AD.

We would like to thank the reviewer for pointing this out, we have corrected this in the paper. (Table 1).

4. Need to define selection process more clearly, such as a). which years were considered eligible EMR entries, b). demographic properties of the main EMR cohorts, c). the sampling proportions based on the imposed criteria, d). the exclusion criteria imposed.

Thank you for your comment. We defined the selection process in the Patient Identification section of Online Methods, which have now been moved into the body of the manuscript (Page 20). We performed minimal exclusion in order to take into account a wider variety of patients. Based on the above suggestion, we have provided more details on the general information and demographic properties of the general EMR population on page 4 and Table 1.

5. It would be good to perform sensitivity analyses to include the temporal ordering of the diagnoses. For example, restricting the comorbidities ascertained only before the AD diagnoses were made while the controls matched with the event occurred as the risk set.

We would like to thank the reviewer for this suggestion. This is a great idea and longitudinal modeling is something that we are actively working on in the context of this cohort especially when carrying out prediction, however we believe this analysis beyond the scope of this current study. We have included this suggestion in our limitations and future directions on page 18.

REVIEWERS' COMMENTS

Reviewer #1 (Remarks to the Author):

The authors have addressed accurately all the reviewers' comments. I have no further comment

Reviewer #3 (Remarks to the Author):

The authors have addressed majority of my concerns through further explanations and clarifications in both responses and updated manuscripts. Although there are still some limitations, I don't see major flaws that would warrant further revision or prohibit publication.

REVIEWERS' COMMENTS

Reviewer #1 (Remarks to the Author):

The authors have addressed accurately all the reviewers' comments. I have no further comment .

Thank you for reviewing the revisions.

Reviewer #3 (Remarks to the Author):

The authors have addressed majority of my concerns through further explanations and clarifications in both responses and updated manuscripts. Although there are still some limitations, I don't see major flaws that would warrant further revision or prohibit publication.

Thank you for reviewing the revisions.